# Perceptions of complementary, alternative, and integrative medicine: A global cross-sectional survey of cardiology researchers and clinicians

Jeremy Y. Ng [1,2]*, Mehvish Masood [1,2], Sivany Kathir [1,2], Holger Cramer [1,2]

1 Institute of General Practice and Interprofessional Care, University Hospital Tübingen, Tübingen, Germany, 2 Robert Bosch Center for Integrative Medicine and Health, Bosch Health Campus, Stuttgart, Germany

* ngjy2@mcmaster.ca, jeremy.ng@med.uni-tuebingen.de

## Abstract

### Background

Complementary, alternative, and integrative medicine (CAIM) has been increasing in popularity among patients with cardiovascular illnesses. However, little is known about perceptions of CAIM among cardiology researchers and clinicians. In response, this study aimed to assess the practices, perceptions, and attitudes towards CAIM among cardiology researchers and clinicians.

### Methods

An anonymous, digital cross-sectional survey was administered to researchers and clinicians who have published articles in cardiology journals indexed in OVID MEDLINE. The survey was sent to 37,915 researchers and clinicians and included 5-point Likert scales, multiple-choice questions, and open-ended questions. Basic descriptive statistics were drawn from quantitative data, and a thematic content analysis was conducted to analyze open-ended responses.

### Results

Among the 309 respondents, the majority (n = 173, 55.99%) identified themselves as both researchers and clinicians in the field of cardiology. While 45.78% (n = 114) of participants expressed agreement regarding the safety of CAIM therapies, 44.40% (n = 111) disagreed on their efficacy. Most respondents believed in the value of conducting research on CAIM therapies (79.2%, n = 198). Respondents perceived mind-body therapies (57.61%, n = 159) and biologically based practices (47.46%, n = 131) as the most promising interventions for the prevention and treatment of cardiovascular conditions. Biofield therapies were the least favoured for integration into mainstream medical practices (11.93%, n = 29).

**Data availability statement:** All data and materials associated with this study are contained within this manuscript or have been posted on the Open Science Framework and can be found here: https://doi.org/10.17605/OSF.IO/MQBW6.

**Funding:** The author(s) received no specific funding for this work.

**Competing interests:** The authors have declared that no competing interests exist.

**Abbreviations:** CAIM, complementary; alternative; and integrative medicine; CAM, complementary and alternative medicine; CVD, cardiovascular disease; MEDLINE, Medical Literature Analysis and Retrieval System Online; NLM, National Library of Medicine; OSF, Open Science Framework; PMIDs, PubMed Identifiers; STROBE, Strengthening the Reporting of Observational Studies in Epidemiology.

## Conclusions

While cardiology researchers and clinicians perceive CAIM therapies to have potential, many are hesitant about integrating such interventions into the current medical system due to a perceived lack of scientific evidence and standardized products. Insights from this study may help establish educational resources for healthcare practitioners.

## Introduction

Cardiology is the medical specialty that focuses on the analysis, diagnosis, and treatment of disorders related to the heart and the circulatory system. Cardiology researchers and clinicians investigate a wide range of conditions, including congenital heart defects, cardiac inflammation (e.g., myocarditis), and cardiovascular diseases (CVDs; e.g., coronary heart disease and peripheral arterial disease) [1–3]. CVDs have garnered heightened attention because of their status as the leading cause of deaths worldwide, accounting for approximately half of noncommunicable disease mortalities and over 17.9 million deaths each year [4–6]. Similarly, congenital heart disease is the leading cause of infant death due to congenital defects worldwide, afflicting approximately 0.8% of live births [7,8]. In response to these critical health challenges, numerous advancements in conventional medicine have emerged. This includes developments in medications (e.g., beta-blockers, angiotensin-converting enzyme inhibitors), surgery (e.g., coronary bypass, aortic valve replacements), nanotechnology, and cardiac rehabilitation programs [9–14]. However, while proven to be effective, the significant emotional, physical, and financial toll of these therapeutics often makes patients increasingly adverse to their usage [15–18]. Medications, for instance, have been reported to have adverse side effects such as rhabdomyolysis, angioedema, upper respiratory symptoms, and hyperkalaemia [18–21]. Similarly, surgeries such as coronary artery bypass grafts and bariatric surgery are known to be both expensive and inaccessible, especially in countries including the United States and China [22,23]. Furthermore, patients have been reported to be in search of therapeutics that address not only the physical and biochemical manifestations of illness but also the emotional, social, and spiritual context of disease [24,25]. As a consequence, patients may view complementary, alternative, and integrative medicine (CAIM) as a viable therapeutic alternative.

According to the US National Center for Complementary and Integrative Health, CAIM is characterized by a diverse array of non-mainstream practices that work *together with* (i.e., complementary approaches), *in place of* (i.e., alternative approaches), or *in coordination with* (i.e., integrative approaches) conventional medicine [26,27]. This involves, but is not limited to, the use of mind-body therapies (e.g., meditation, biofeedback, yoga), biologically based practices (e.g., vitamins, botanicals, special foods and diets), manipulative and body-based practices (e.g., massages, reflexology, chiropractic therapy), and whole medical systems (e.g., traditional Chinese medicine, Ayurveda, naturopathic medicine) [27]. Prior research has

shown that an overwhelming number of individuals partake in CAIM practices. A systematic review of national studies published from 2010 to 2019 found that the reported prevalence of traditional, complementary, and alternative medicine use within the general population ranged from 24 to 71.3% [28]. Wood et al. found that 64% of 107 CVD patients randomly selected from a stratified cohort of 2,487 eligible participants identified themselves as using complementary and alternative medicines (CAM) such as nutritional supplements and megadose vitamins [29]. This usage suggests that understanding such practices is necessary for aiding patients in cardiovascular settings.

Unfortunately, the polarizing nature of CAIM has limited research into its safety and usage [30–33]. Specifically, CAIM has been perceived as a challenge to the traditional scientific training and expertise of many physicians and researchers, which may prove to be harmful to patient outcomes [31–33]. Insufficient training and insight regarding CAIM have been reported to cause healthcare staff to be ill-equipped and hesitant to provide resources or insight when approached by patients [25,31–34]. Moreover, given that many patients rely on healthcare systems and physician appointments for information regarding treatments, providing unclear information may result in patients turning to sources that may not be credible or present ambiguous information [35,36]. These potentially adverse consequences underscore the importance of addressing gaps in CAIM-based knowledge for researchers and clinicians.

To the best of our knowledge, research has not been conducted on the perspectives of CAIM among cardiology researchers and physicians specifically, with previous literature focusing on CAIM perspectives *in general* [37], or focused specifically on perspectives of patients with cardiovascular disease [38,39]. This has hindered the ability to determine whether CAIM-based resources are required for cardiology researchers and clinicians and, if so, what specific issues need to be addressed. In response to this gap in knowledge, this international, cross-sectional survey aims to understand how cardiology researchers and clinicians perceive CAIM. This exploration is intended to provide a more comprehensive insight into global cardiology perspectives regarding CAIM by delving into the foundations of reservations and support for such practices and exploring the mechanisms by which such views may be addressed. This study is descriptive, and there were no formal hypotheses for this study.

## Methods

### Transparency statement

This study received approval from the University Tübingen Research Ethics board before commencement (REB Number: 389/2023BO2). Implied consent was collected from each participant; upon clicking the survey invitation link, participants were presented with the consent form, and were informed that "By completing the survey your consent to participate is implied".

The study was registered with the Open Science Framework (OSF) and can be found here: https://doi.org/10.17605/OSF.IO/FTNMD. The protocol, along with study material and raw data, can also be found on OSF at https://doi.org/10.17605/OSF.IO/MQBW6 [40]. We followed the Strengthening the Reporting of Observational Studies in Epidemiology (STROBE) Statement for cross-sectional studies in reporting our findings [41].

### Study design

This study was an anonymous, cross-sectional online survey administered to authors who have published in cardiology medical journals indexed on OVID MEDLINE [42].

### Sampling framework

A sample of corresponding authors from all articles published between December 15, 2018 and August 1, 2023 was selected from a sample of cardiology journals found at https://journal-reports.nlm.nih.gov/broad-subjects/ [43]. Authors with published articles of any type were included in this study. The search strategy was based on National Library of

Medicine (NLM) IDs extracted from selected journals and was run on OVID MEDLINE. The resulting list of approximately 150,000 PubMed Identifiers (PMIDs) was exported as a.csv file and input into an R script (built based on the easyPubMed package) [44]. This retrieved the authors' names, affiliation institutions, and email addresses that were used in this study. A power analysis was not included because this is a convenience sample with descriptive work that lacks any inferential testing. The search strategy can be found at https://osf.io/59wxg.

### Participant recruitment

Prospective participants identified from the sampling framework were the only individuals contacted to partake in this study and complete the closed survey. To be eligible, participants needed to be able to read and write in English and self-identify as a researcher and/or clinician within the field of cardiology. Respondents were not required to have any background or understanding of CAIM because this study intended to capture how CAIM is perceived within the general cardiology community and whether there are gaps in knowledge that may impact its integration into research/clinical practice. Emails based on a recruitment script that included an explanation of the study and its goals, along with a link to the survey, were sent to prospective participants using SurveyMonkey [45]. When the survey link was clicked, participants could review the informed consent form, which included the purpose of the study, the length of time to complete the survey, the principal investigator's name, and data storage information. Participants were then asked to respond to a yes/no question asking for consent to partake in the study. If participants responded 'yes' (i.e., they gave consent), they were given access to the first page of the online survey. Except for this consent question, participants were able to skip any questions to which they did not wish to respond. All responses were collected anonymously. The email list was expected to contain duplicates (i.e., authors corresponding to multiple manuscripts with the sample set), which were removed before sending out the survey. Non-functioning or invalid email addresses were also accounted for within SurveyMonkey. The survey was open between November 1, 2023 and December 26, 2023. Participants were sent reminder emails in the first, second, and third weeks following the original invitation email. The survey was voluntary, and respondents were not provided with financial compensation for participation.

### Survey design

The SurveyMonkey platform was used to create and distribute the survey [45]. Following a screening question, participants were asked a series of demographic questions (7 questions), followed by a section about their perceptions of CAIM (25 questions). Most of the survey questions used a multiple-choice and 5-point Likert scale (i.e., strongly disagree, disagree, neither agree nor disagree, agree, strongly agree) format. Some survey items asked respondents about their views on CAIM therapies *in general*, along with their opinions on five subcategories of CAIM therapies: 1) mind-body therapies (e.g., meditation, biofeedback, hypnosis, yoga, tai chi, imagery, creative outlets); 2) biologically based practices (e.g., vitamins and dietary supplements, botanicals, special foods and diets, phytotherapy); 3) manipulative and body-based practices (e.g., massage, chiropractic therapy, reflexology); 4) biofield therapies (e.g., reiki, therapeutic touch); and 5) whole medical systems (e.g., Ayurvedic medicine, traditional Chinese medicine, acupuncture, naturopathic medicine). A small minority of questions were open-ended. Depending on the context, some questions were also solely presented to participants who identified as researchers or clinicians. The first iteration of the survey was pilot tested by researchers not involved in this study. Feedback from the pilot-test was reviewed and incorporated into the survey prior to administration. The survey can be found at https://osf.io/ef8hx.

### Data management and analysis

Quantitative data was used to generate basic descriptive statistics, including counts and percentages. For the narrative summary, participants that had 'agreed' or 'strongly agreed' with a statement on the 5-point Likert scale were grouped together as supporting/agreeing with that statement and respondents that 'disagreed' or 'strongly disagreed' were grouped

together as opposing/disagreeing with that statement. Qualitative data was analysed using a thematic content analysis [46]. Two research team members (JYN, SK) independently coded participant responses, followed by iterative rounds of discussion (JYN, SK) to reach a final consensus on the codes and themes. The data was then categorically classified into distinct tables for reporting.

## Results

Out of the 37,915 email invitations, 10,543 remained unopened and 4,872 bounced. A total of 309 respondents met the eligibility criteria and took part in the survey (0.82% response rate for all emails sent; 1.37% response rate for all emails opened). Of the sent invitations, the survey completion rate was 71%, and the average completion time was 8 minutes 11 seconds. All survey questions were optional, resulting in a varied response rate for each question. Anonymous raw survey responses can be found at https://osf.io/9e4yt.

### Demographic characteristics

Most participants self-identified as male (67.88%, n = 205) and between 35 and 54 years old (53.31%, n = 161). A total of 38.19% (n = 118) of participants identified solely as researchers, 5.83% (n = 18) identified solely as clinicians, and 55.99% (n = 173) identified as *both* researchers and clinicians within the cardiology field. According to the World Health Organization World Regions classification [47], respondents were predominantly located in Europe (46.15%, n = 138) and the Americas (33.44%, n = 100). In terms of professional roles, most respondents identified as 'faculty members and/or principal investigators' (54.97%, n = 166), 'clinicians' (45.70%, n = 138), or 'scientists in academia' (27.15%, n = 82). Many participants were senior researchers or clinicians with over 10 years of experience post formal career education (64.45%, n = 194). The primary research areas for most participants was clinical (83.27%, n = 219) and epidemiological research (21.67%, n = 57). Most respondents had never conducted research in the field of CAIM (77.01%, n = 201). Complete participant demographics are provided in Table 1.

### CAIM perceptions

Mind-body therapies (57.61%, n = 159) and biologically based practices (47.46%, n = 131) were considered to be the most promising interventions for the prevention, treatment, and/or management of cardiovascular diseases and conditions. When asked about the safety and effectiveness of CAIM therapies overall, most participants agreed on its safety (45.78%, n = 114), but disagreed on its effectiveness (44.40%, n = 111).

 Views regarding the safety and efficacy of specific CAIM subcategories were dependent on the modality presented. A total of 72.07% (n = 178) believed mind-body therapies were safe. In contrast, approximately one-third of participants agreed with the safety of: biologically based practices (33.74%, n = 83), manipulative and body-based practices (36.74%, n = 90), biofield therapies (40.75%, n = 99), and whole medical systems (32.58%, n = 79; Fig 1). Among all the CAIM modalities, mind-body therapies were most often viewed as effective (38.87%, n = 96). In comparison, biologically based practices, biofield therapies, and whole medical systems garnered variable levels of support, with 9.87% to 23.77% of participants believing that such modalities are effective and 31.56% to 44.67% disagreeing with their effectiveness (Fig 2).

 Participants were asked about the perceived benefits associated with CAIM. Several key advantages were acknowledged: 'focus on prevention and lifestyle changes' (60.42%, n = 145), 'holistic approach to health and wellness' (58.33%, n = 140), 'expanded treatment options for patients' (56.25%, n = 135), 'empowerment of patients to take control of their health' (52.92%, n = 127), and 'increased patient satisfaction and well-being' (50.42%, n = 121; Fig 3). Participants were also asked about the perceived challenges they associate with CAIM. The majority of clinicians had concerns with the 'lack of scientific evidence for safety and efficacy' (92.65%, n = 227), 'lack of standardization in product quality and dosing' (86.53%, n = 212), 'limited regulation and oversight' (74.29%, n = 182), 'difficulty in distinguishing legitimate practices from

**Table 1. Characteristics of survey participants.**

| Demographic Factors | Participant Characteristics | n (%) |
|---|---|---|
| Sex (n = 302) | Female | 94 (31.13%) |
| | Male | 205 (67.88%) |
| | Intersex | 0 (0.00%) |
| | Prefer not to say | 2 (0.66%) |
| | Prefer to self-describe | 1 (0.33%) |
| Age (years) (n = 302) | Under 18 | 0 (0.00%) |
| | 18-24 | 1 (0.33%) |
| | 25-34 | 39 (12.91%) |
| | 35-44 | 78 (25.83%) |
| | 45-54 | 83 (27.48%) |
| | 55-64 | 59 (19.54%) |
| | 65 or older | 40 (13.25%) |
| | Prefer not to say | 2 (0.66%) |
| Visible Minority Status (n = 302) | Yes | 37 (12.25%) |
| | No | 250 (82.78%) |
| | Prefer not to say | 15 (4.97%) |
| World Region (n = 299) | Africa | 6 (2.01%) |
| | America | 100 (33.44%) |
| | Eastern Mediterranean | 11 (3.68%) |
| | Europe | 138 (46.15%) |
| | South-East Asia | 24 (8.03%) |
| | Western Pacific | 16 (5.35%) |
| | Prefer not to say | 4 (1.34%) |
| Current Position (n = 302) | Clinician Student | 6 (1.99%) |
| | Clinician | 138 (45.70%) |
| | Graduate student | 10 (3.31%) |
| | Postdoctoral fellow | 12 (3.97%) |
| | Faculty member/Principal Investigator | 166 (54.97%) |
| | Research support staff | 11 (3.64%) |
| | Scientist in academia | 82 (27.15%) |
| | Scientist in industry | 2 (0.66%) |
| | Scientist in third sector | 4 (1.32%) |
| | Government scientist | 4 (1.32%) |
| | Other (please specify) | 9 (2.98%) |
| Career Stage (n = 301) | Graduate or clinician student | 10 (3.32%) |
| | Early career (<5 yrs post formal education) | 40 (13.29%) |
| | Mid-career (5–10 yrs post formal education) | 57 (18.94%) |
| | Senior (>10 yrs post formal education) | 194 (64.45%) |
| Primary Research Area (n = 263) | Clinical research | 219 (83.27%) |
| | Preclinical research – in vivo | 47 (17.87%) |
| | Preclinical research – in vitro | 33 (12.55%) |
| | Health systems research | 23 (8.75%) |
| | Health services research | 31 (11.79%) |
| | Methods research | 20 (7.60%) |
| | Epidemiological research | 57 (21.67%) |
| | Other (please specify) | 7 (2.66%) |

*(Continued)*

**Table 1.** (Continued)

| Demographic Factors | Participant Characteristics | n (%) |
|---|---|---|
| Area of CAIM Research Experience (n = 261) | Mind-body therapies | 13 (4.98%) |
| | Biologically based practices | 42 (16.09%) |
| | Manipulative and body-based practices | 5 (1.92%) |
| | Biofield therapies | 2 (0.77%) |
| | Whole medical systems | 11 (4.21%) |
| | I have never conducted any CAIM research | 201 (77.01%) |
| | Other (please specify) | 3 (1.15%) |

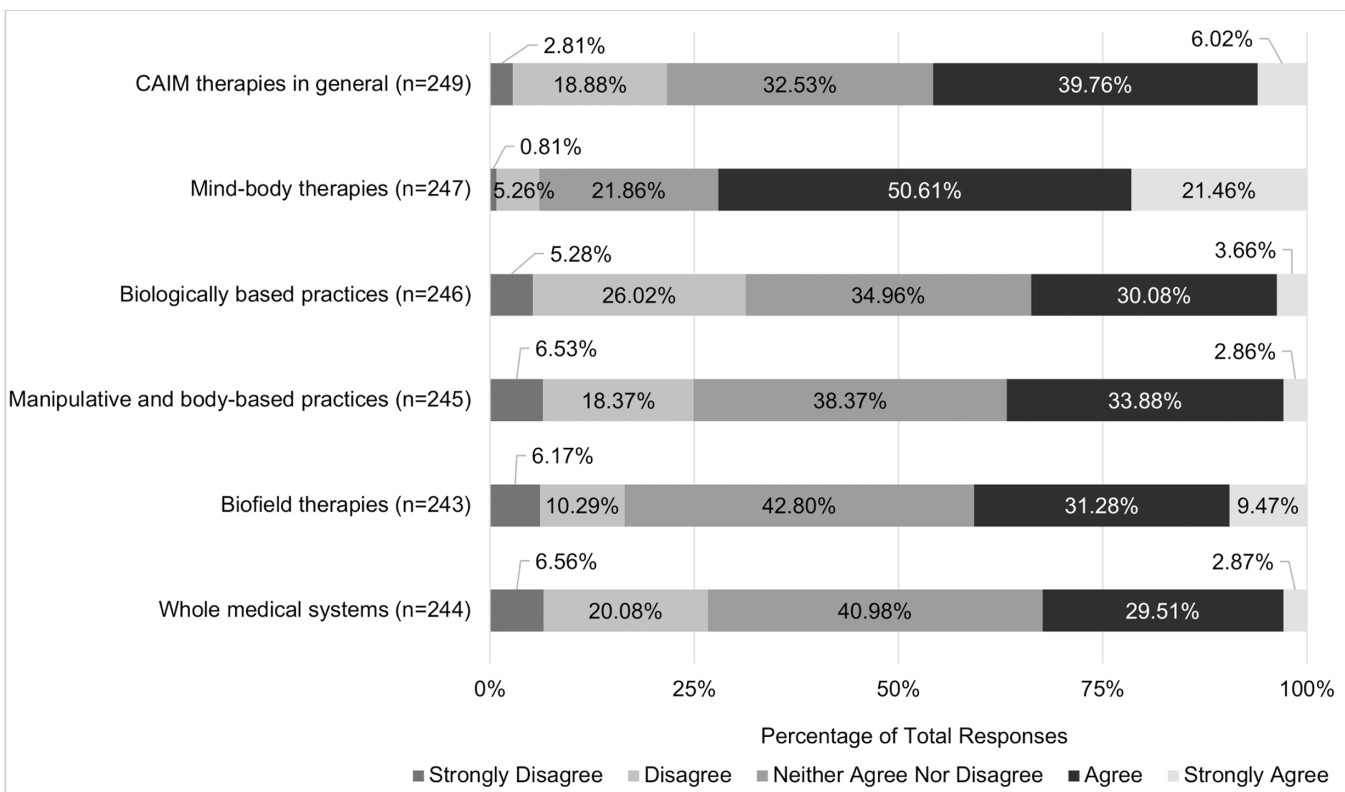

**Fig 1. Participants' Agreement Expressed Towards the Safety of Each Complementary, Alternative, and Integrative Medicine (CAIM) Category.**

scams or fraudulent claims' (68.57%, n = 168), and the 'limited integration with mainstream healthcare systems' (50.61%, n = 124; Fig 4).

## Clinical experiences with CAIM

Participants reported that biologically based practices (75.58%, n = 130) and mind-body therapies (54.65%, n = 94) were the most sought out by patients through counselling or disclosed usage. Most participants (57.06%, n = 97) reported that 0–10% of their patients disclose using and/or seek counselling for CAIM therapies. This was followed by 25.88% (n = 44) indicating that 11–20% of patients do so. A substantial portion of participants acknowledged being

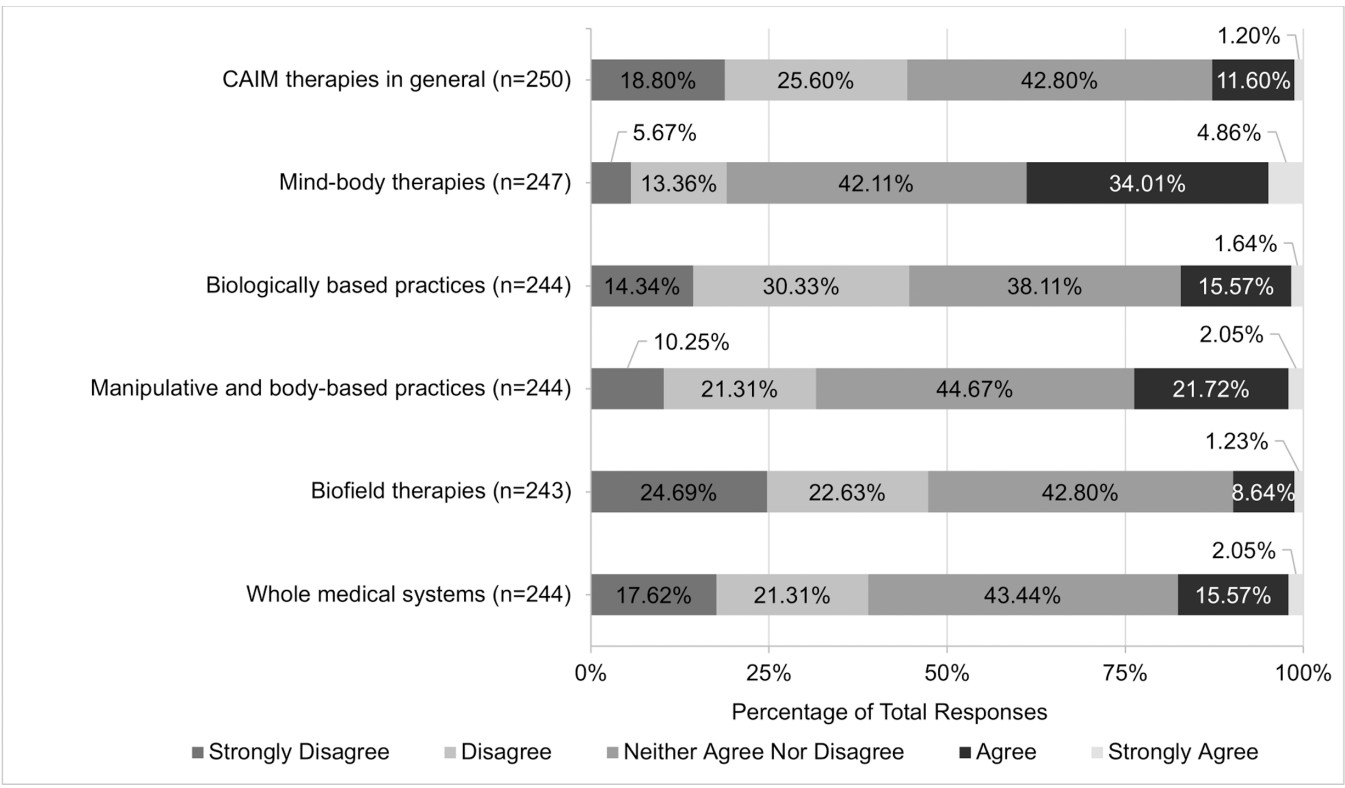

**Fig 2. Participants' Agreement Expressed Towards the Efficacy of Each Complementary, Alternative, and Integrative Medicine (CAIM) Category.**

asked 'occasionally' (58.84%, n = 163) and 'often' (13.00%, n = 36) about CAIM by individuals outside of research and/ or clinical settings (e.g., family members, friends). Mind-body therapies (44.44%, n = 76) and biologically based practices (29.24%, n = 50) were reported as the areas of CAIM that the greatest proportion of participants had practiced or recommended to patients. A large portion (37.43%, n = 64) also indicated that they have never used or recommended CAIM therapies.

The majority of respondents expressed discomfort in counselling (47.06%, n = 72; Fig 5) and recommending (59.86%, n = 91; Fig 6) CAIM therapies to patients in general. Comfort with types of CAIM therapeutics varied by subcategory. Participants were the most uncomfortable with counselling (59.18%, n = 87) and recommending (69.83%, n = 102) biofield therapies to patients. Conversely, respondents were the least uncomfortable with counselling (42.00%, n = 63) and recommending (36.91%, n = 55) mind-body therapies to patients.

When participants were surveyed about whether CAIM therapies should be covered by insurance, the majority (42.00%, n = 105) expressed disagreement. This sentiment was echoed across various CAIM subcategories, including biologically based practices (46.75%, n = 115), manipulative and body-based practices (40.81%, n = 100), biofield therapies (53.91%, n = 131), and whole medical systems (46.72%, n = 114). Interestingly, opinions regarding insurance coverage for mind-body therapies were more evenly distributed, with 30.89% (n = 76) in support, 36.18% (n = 89) as neutral, and 32.92% (n = 81) being opposed. Similarly, with the exception of mind-body therapies, which had relatively high rates of supportive (34.42%, n = 85) and neutral (36.44%, n = 90) responses, the majority of participants disagreed with the integration of CAIM into mainstream medical practice, with responses ranging from 42.04% to 52.67% in disagreement depending on the CAIM subcategory (Fig 7).

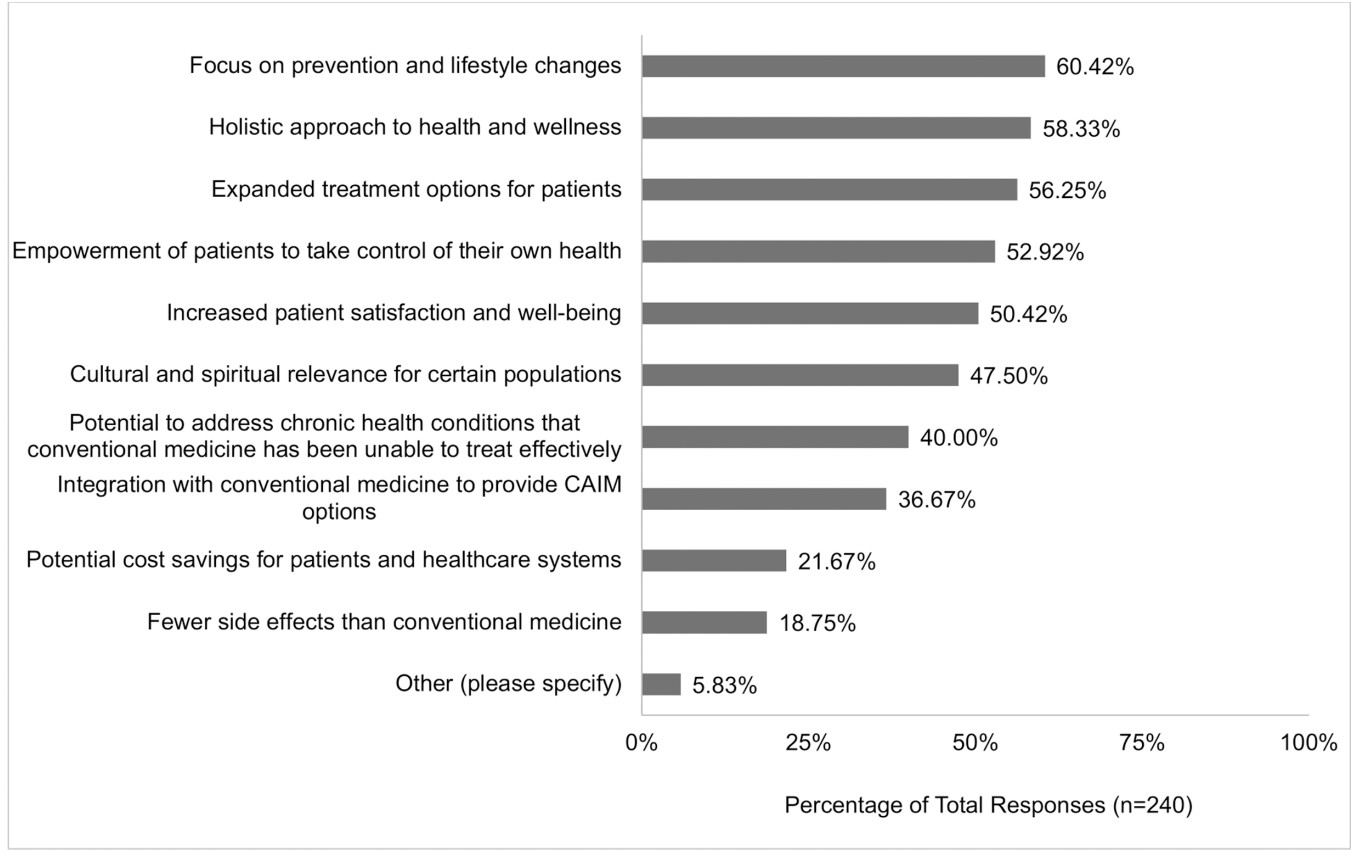

**Fig 3. Benefits Participants Perceive as Associated with Complementary, Alternative, and Integrative Medicine (CAIM).**

## Attitudes towards CAIM research

Most participants 'agreed' or 'strongly agreed' that there is value in research being conducted on CAIM therapies (79.20%, n = 198; Fig 8). Specifically, high levels of agreement were observed when asked about the value in conducting further research on mind-body therapies (78.95%, n = 195), biologically based practices (75.92%, n = 186), manipulative and body-based practices (58.61%, n = 143), and whole medical systems (59.42%, n = 145). The only CAIM modality with varied opinions on conducting research was biofield therapies, with similar levels of support (36.22%, n = 88), neutrality (33.33%, n = 81), and disagreement (30.45%, n = 74) being observed. Similarly, most participants agreed that more research funding should be allocated to study CAIM therapies in general (52.42%, n = 130). In terms of specific CAIM modalities, mind-body therapies (53.47%, n = 131), biologically based practices (75.92%, n = 186), manipulative and body-based practices (42.04%, n = 103), and whole medical systems (45.09%, n = 110) had general agreement that more research funds should be allocated to study the respective CAIM categories. The modality with the smallest proportion of participants supporting increased research funds was biofield therapies (23.87%, n = 58).

## CAIM education

Most respondents reported never receiving formal education (80.70%, n = 138; Fig 9) or supplementary education (67.06%, n = 114; Fig 10) on the use of CAIM therapies in general. However, a portion had received supplementary

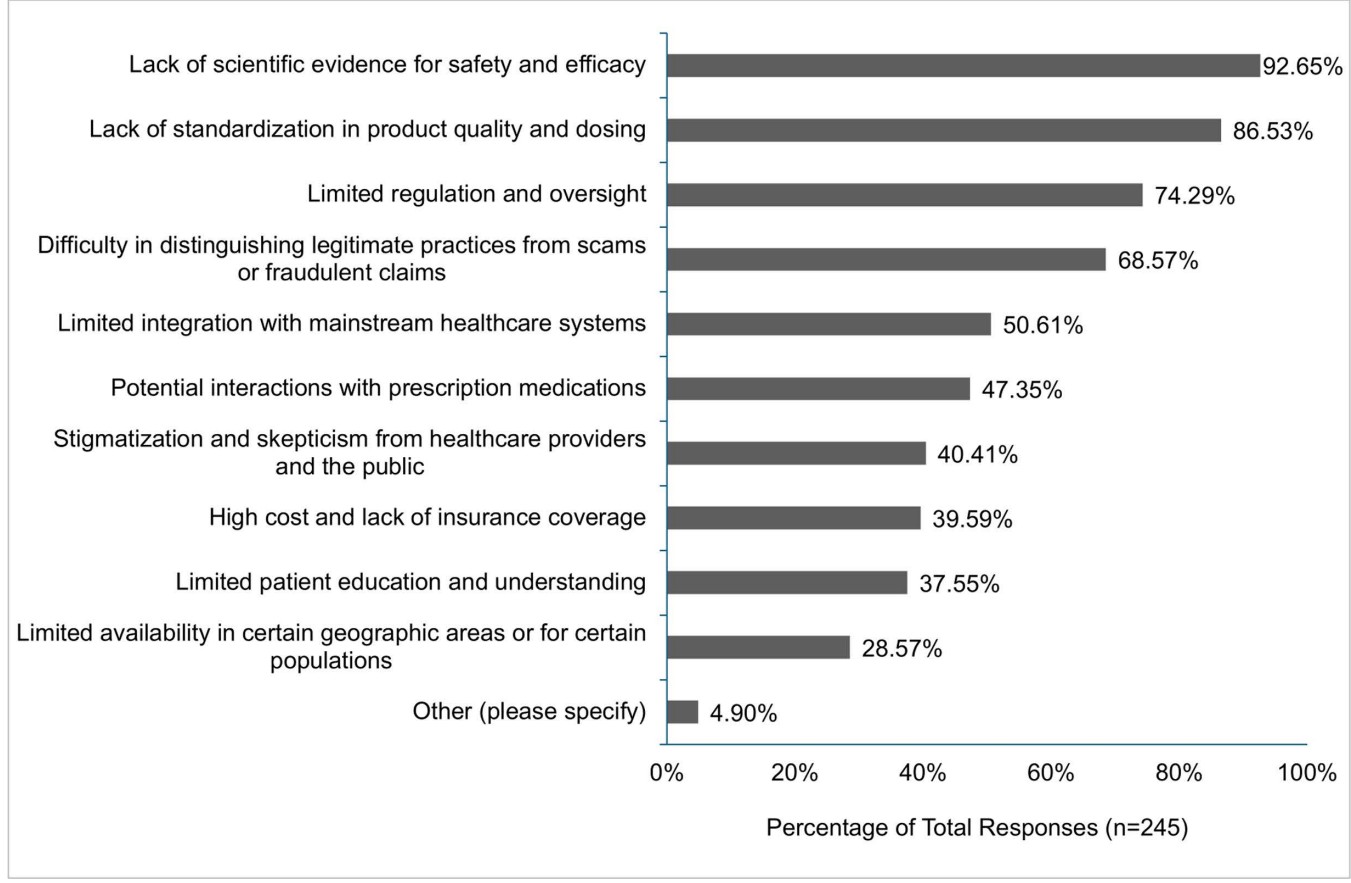

**Fig 4. Challenges Participants Perceive as Associated with Complementary, Alternative, and Integrative Medicine (CAIM).**

education on mind-body therapies (18.24%, n = 31) and biologically based practices (17.06%, n = 29). Many participants support clinicians receiving more formal (53.20%, n = 133) and supplementary education (62.10%, n = 154) on CAIM therapies. In particular, similar rates of support for additional formal and supplementary education, respectively, were observed across most CAIM subcategories, including mind-body therapies (51.02%, n = 126; 57.72%, n = 142), biologically based practices (55.11%, n = 135; 54.06%, n = 133), manipulative and body-based therapies (37.04%, n = 90; 41.56%, n = 101), and whole medical systems (33.2%, n = 81; 35.39%, n = 86). Biofield therapies was the sole exception, with limited participants supporting the necessity for additional formal (22.31%, n = 54) and supplementary (24.69%, n = 60) training. When asked where participants would seek out information if they wanted to learn more about CAIM, most respondents indicated they would refer to 'academic literature' (90.22%, n = 249), 'conference presentations or workshops' (35.14%, n = 97), and 'health information pages on the internet' (30.07%, n = 83).

## Thematic analysis

When participants were asked to share their remaining perceptions of CAIM, six major themes (twenty parent codes) were emerged from our analysis of 55 open-ended survey responses. The first theme, "need for more rigorous research and resources to inform decision making about CAIM" encompasses opinions that advocate for more rigorous research, more education, and the need to standardize CAIM therapies. Comments emphasizing the importance of scientific methods in CAIM research were also placed under this theme. The second theme, "CAIM products can be harmful due

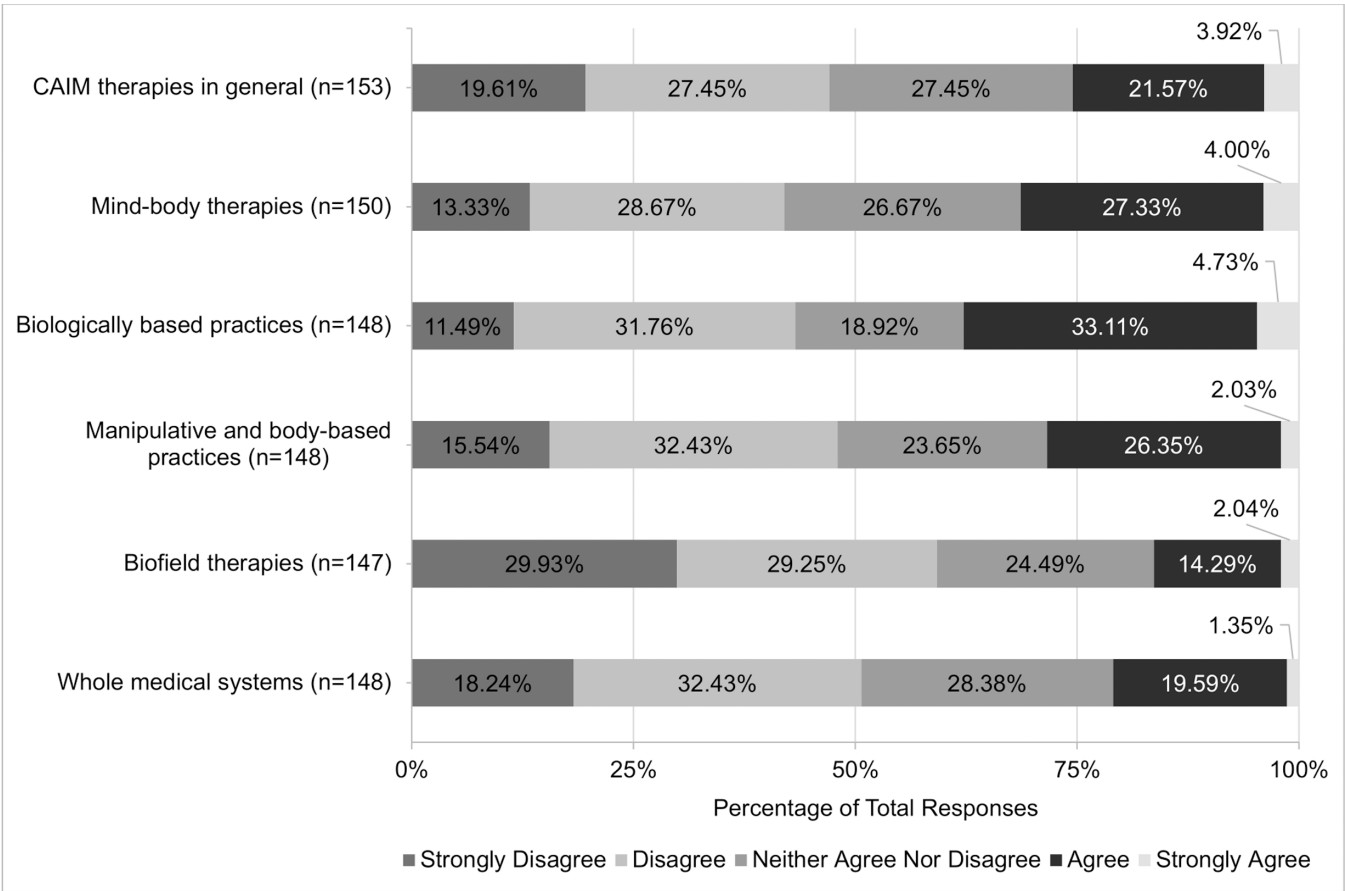

**Fig 5. Clinicians' Agreement Expressed Towards Feeling Comfortable Counselling Patients on Each Complementary, Alternative, and Integrative Medicine (CAIM) Category.**

to misuse/misinformation" includes responses that note concerns with CAIM therapies being used as a replacement for traditional treatment instead of in a complementary fashion. Additionally, comments expressing concern over exploitative marketing techniques were also coded under this theme. The third theme, "concern with CAIM categories" encompasses responses that did not agree with the subcategorization of CAIM therapies. Simultaneously, within the same theme, respondents also expressed concerns with treating CAIM as a homogenous entity. For example, one participant was conflicted regarding the *overall* efficacy of CAIM because they believed in the effectiveness of some CAIM interventions (i.e., relaxation therapy) but not others (i.e., chiropractic manipulations). The next theme, "CAIM is appealing to patients" includes comments that acknowledge the cultural significance of CAIM and how patients practice CAIM more than what is disclosed to healthcare providers. The "polar or personal opinions/experiences (pro vs. anti)" theme encompassed comments that were either in support or against CAIM therapies without justification behind their standpoint. Comments that were personal in nature and/or anecdotal were also categorized under this theme. The last theme, "CAIM in the clinical and/or academic field of cardiology" includes comments that spoke specifically about CAIM in cardiology. Comments under this theme mention the potential of CAIM, the need for more research on CAIM in relation to cardiology, and how certain therapies can be harmful to cardiac conditions (e.g., adverse drug reactions). Coding and thematic analysis data are available at: https://osf.io/nctjd. A summary of the thematic analysis, including representative quotations, is available in Table 2.

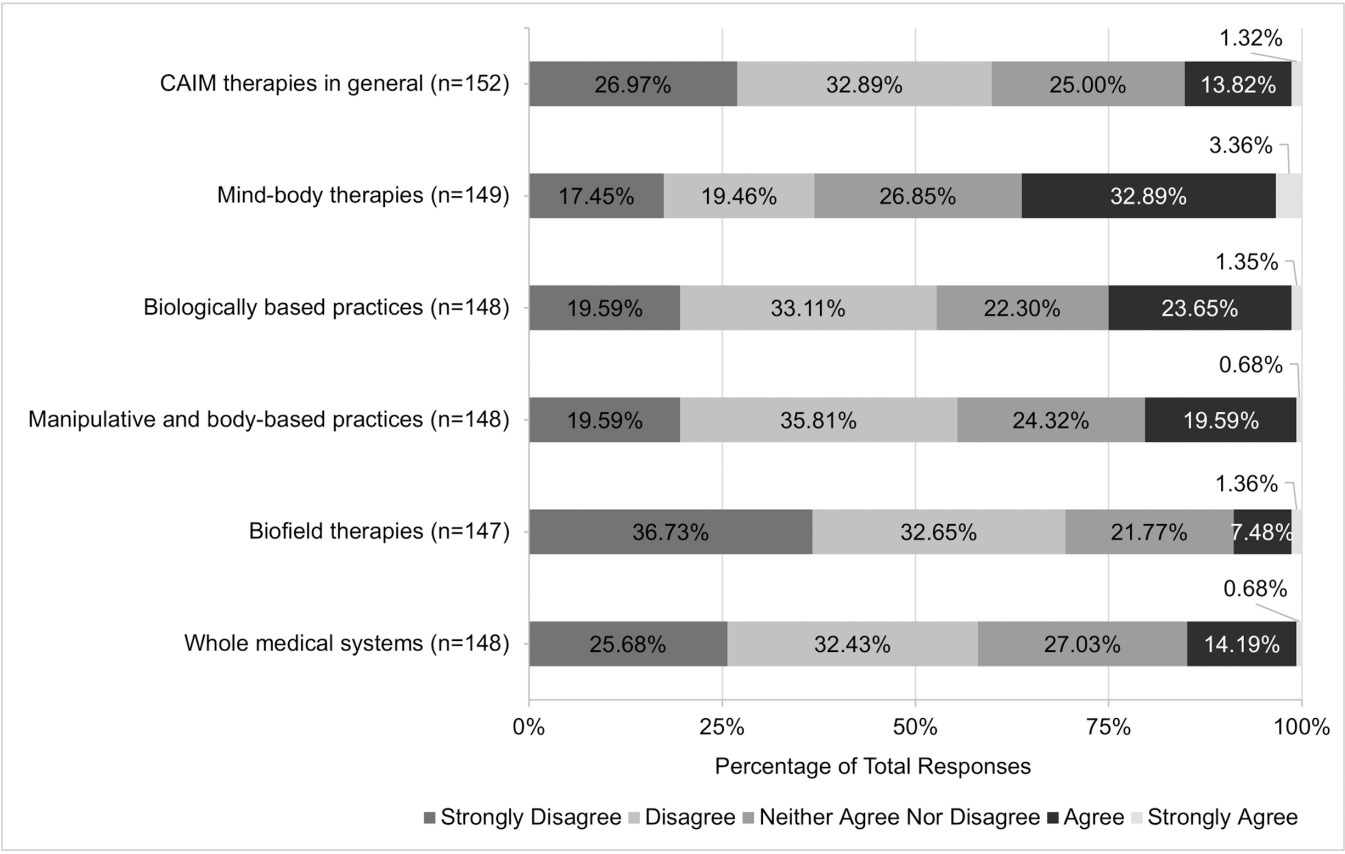

**Fig 6. Clinicians' Agreement Expressed Towards Feeling Comfortable Recommending Each Complementary, Alternative, and Integrative Medicine (CAIM) Category to Patients.**

## Discussion

The objective of this study was to investigate the perceptions and awareness of CAIM practices among cardiology researchers and clinicians. While existing literature predominantly examines patients' perceptions and attitudes towards CAIM [48–51], our study focuses on the perspectives of professionals involved in clinical or academic cardiology specifically. Findings demonstrate that, while researchers and clinicians in the field of cardiology view CAIM therapies as generally safe and potentially beneficial, there is a significant need for more research and educational training on certain CAIM therapies. Given that modifiable lifestyle factors play a significant role in cardiac health, understanding the perceptions of CAIM held by cardiology professionals is imperative [52,53].

### Comparative literature

A large proportion of cardiology researchers and clinicians reported feeling skeptical about the efficacy of CAIM therapies. Survey responses suggest that this may be due to CAIM being perceived as lacking high-quality, rigorous evidence. This sentiment has been reflected in similar studies investigating the perspectives of researchers and clinicians within neurology, psychiatry, and oncology [54–58]. Perceptions of CAIM held by healthcare professionals have been shown to depend on several different factors, including personal experiences, the recommendations of respected peers and case reports about treatments and conditions in CAIM journals [59–61]. It was also found that, while some medical schools do provide

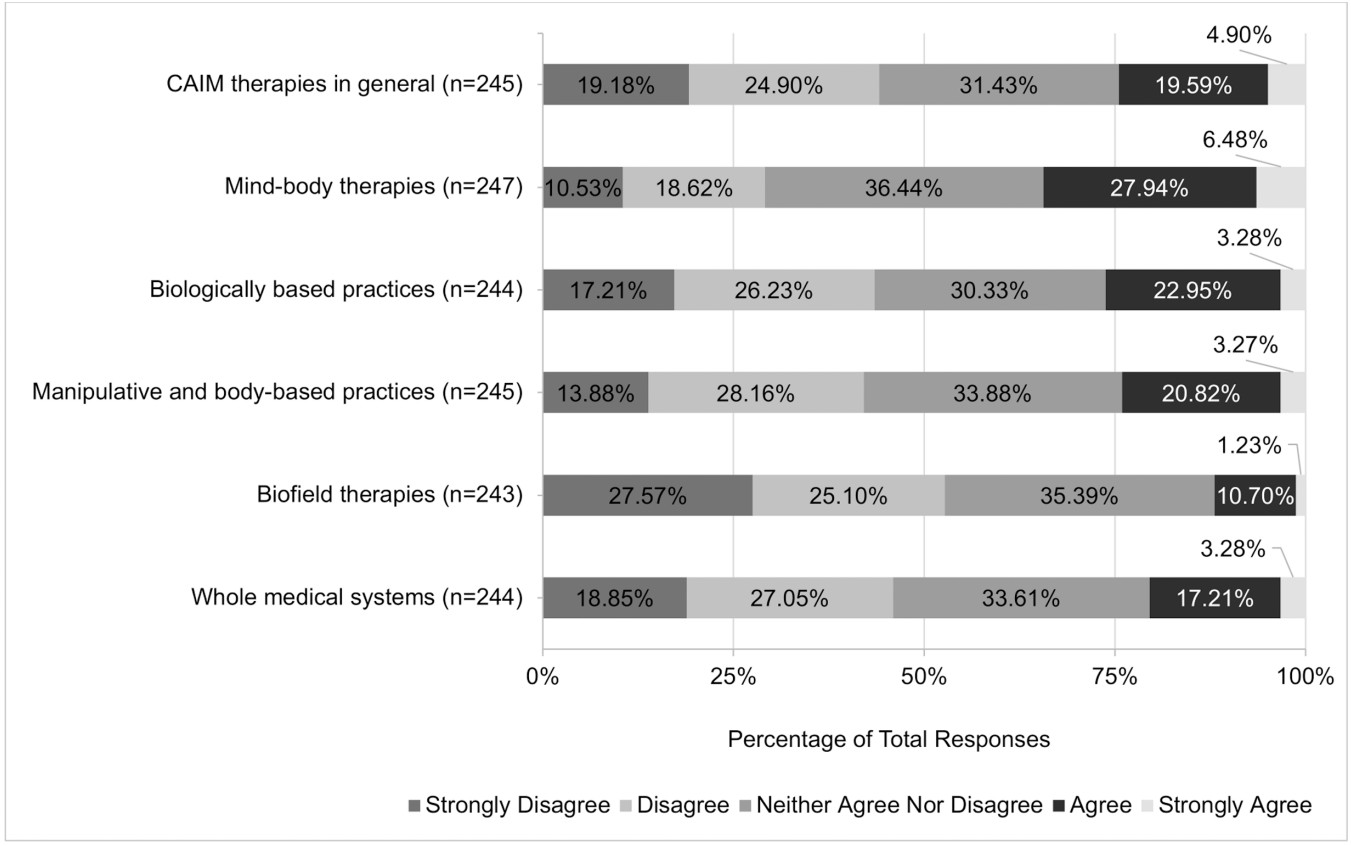

**Fig 7. Participants' Agreement Expressed Towards the Integration of Each Complementary, Alternative, and Integrative Medicine (CAIM) Category into Mainstream Medical Practices.**

education on CAIM, they approach the topic with overt bias (both overly positive or overly negative), which is reported to impact the perspective of medical students and thus their subsequent willingness to refer patients to CAIM therapies [62].

Furthermore, while cardiology researchers and clinicians present a lack of comfort with counseling or recommending CAIM, many report being asked about such therapies by individuals outside of research/clinical settings (e.g., family and friends). While similar trends have been observed within the literature [54–58], such viewpoints are particularly relevant for cardiology due to the strong influence that lifestyle factors play in the management of cardiovascular disease [52,53]. Within integrative cardiology, a subfield of cardiology that focuses on using evidence-based healing modalities that focus on health and prevention, "lifestyle modifications, including nutrition and physical activity, are pillars that are complemented by mind-body interventions, acupuncture, and [the] appropriate use of nutraceuticals" [63]. The significance of lifestyle factors, along with results from the present analysis and the association of cardiovascular disease with the use of CAIM therapies [18,29,64,65], emphasize the importance of understanding cardiology-related perceptions of CAIM.

Participants reported that, of the five CAIM subcategories, mind-body therapies and biologically based practices were viewed as the most promising and were the most sought-out by patients. This is similar to findings in other studies [54–58], likely because these therapies are accessible and generally safe to experiment with [66–68]. Further, mind-body therapies (e.g., yoga, relaxation, variants of cognitive behavioral therapy) are widely viewed as low-cost methods to improve blood pressure, mental health, and overall fitness [69–72]. Several studies note that such therapies help with systemic inflammation, stress, the cardiac autonomic nervous system, and cardiovascular risk factors [72].

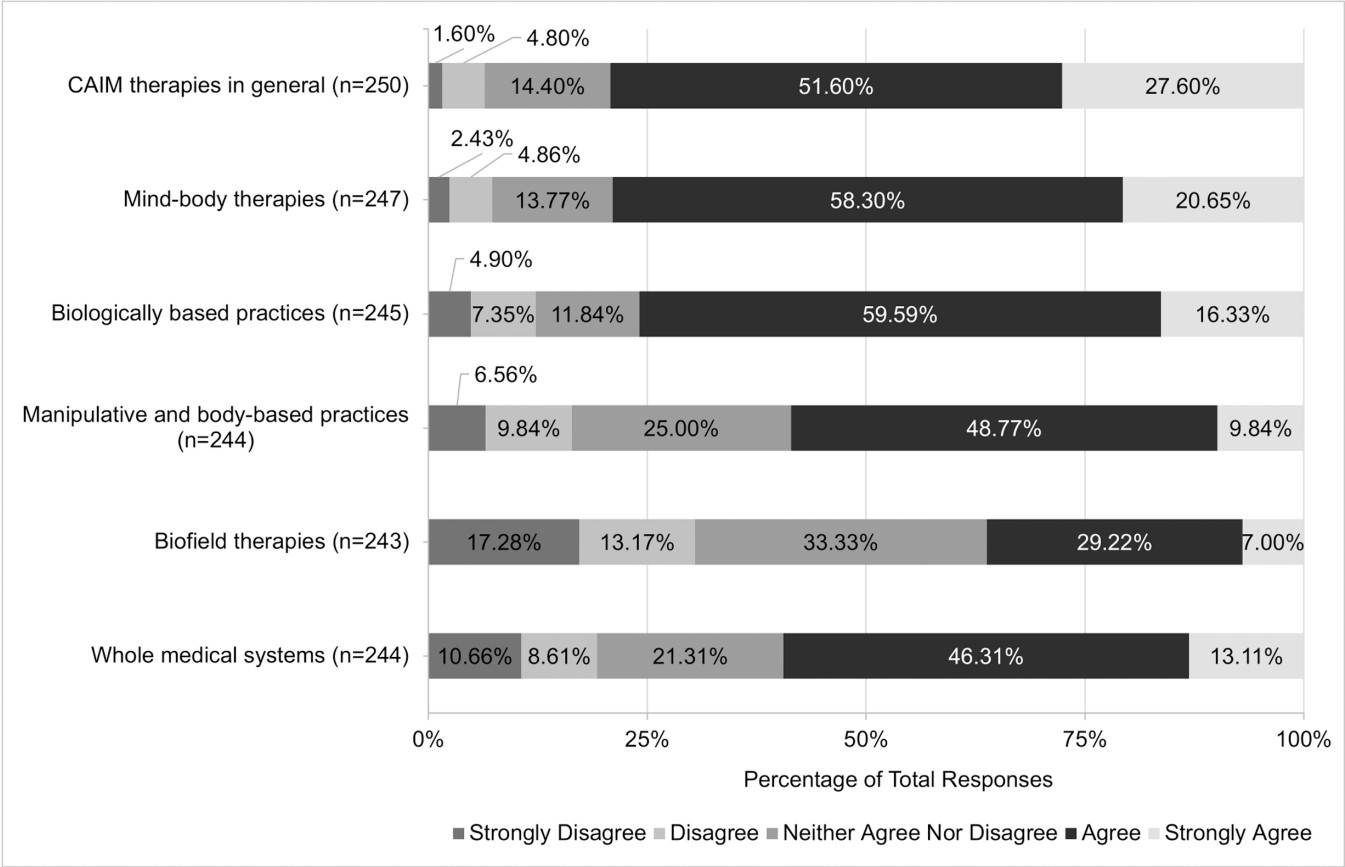

**Fig 8. Participants' Agreement Expressed Towards the Value of Conducting Research on Each Complementary, Alternative, and Integrative Medicine (CAIM) Category.**

While biologically-based factors have been noted to have potential and to be popular among patients [29,64], respondents disagreed on its efficacy. The thematic analysis revealed that participants were hesitant about CAIM due to the potential for treatments to adversely impact and interfere with cardiovascular medications and conditions. For example, bitter orange and ginkgo biloba extracts or supplements, common ingredients in CAIM medications, have been linked to increased heart rate [73–75]. The risk for biologically-based factors, such as herbal supplements, to interfere with cardiovascular medication may be why many cardiology researchers and clinicians have consequently expressed hesitation about its efficacy.

Biofield therapies were viewed the least favorably by cardiology researchers and clinicians. Specifically, participants provided the least support towards enhancing formal and supplementary education and allocating research funding towards this CAIM subcategory. Further, respondents were not comfortable with counselling and recommending these practices to patients or integrating biofield therapies into mainstream medical practices. While shown potential to aid with cardiac conditions [76–78], prior literature has suggested that methodological shortcomings and a lack of research transparency associated with biofield therapy studies have limited confidence in these interventions [79,80]. Respondents within the present analysis identified a 'lack of scientific evidence for safety and efficacy' as the greatest challenge associated with CAIM. Consequently, the lack of rigor and transparency associated with biofield therapy research, as suggested within prior literature [79,80], may be why respondents presented with the most discomfort towards such interventions.

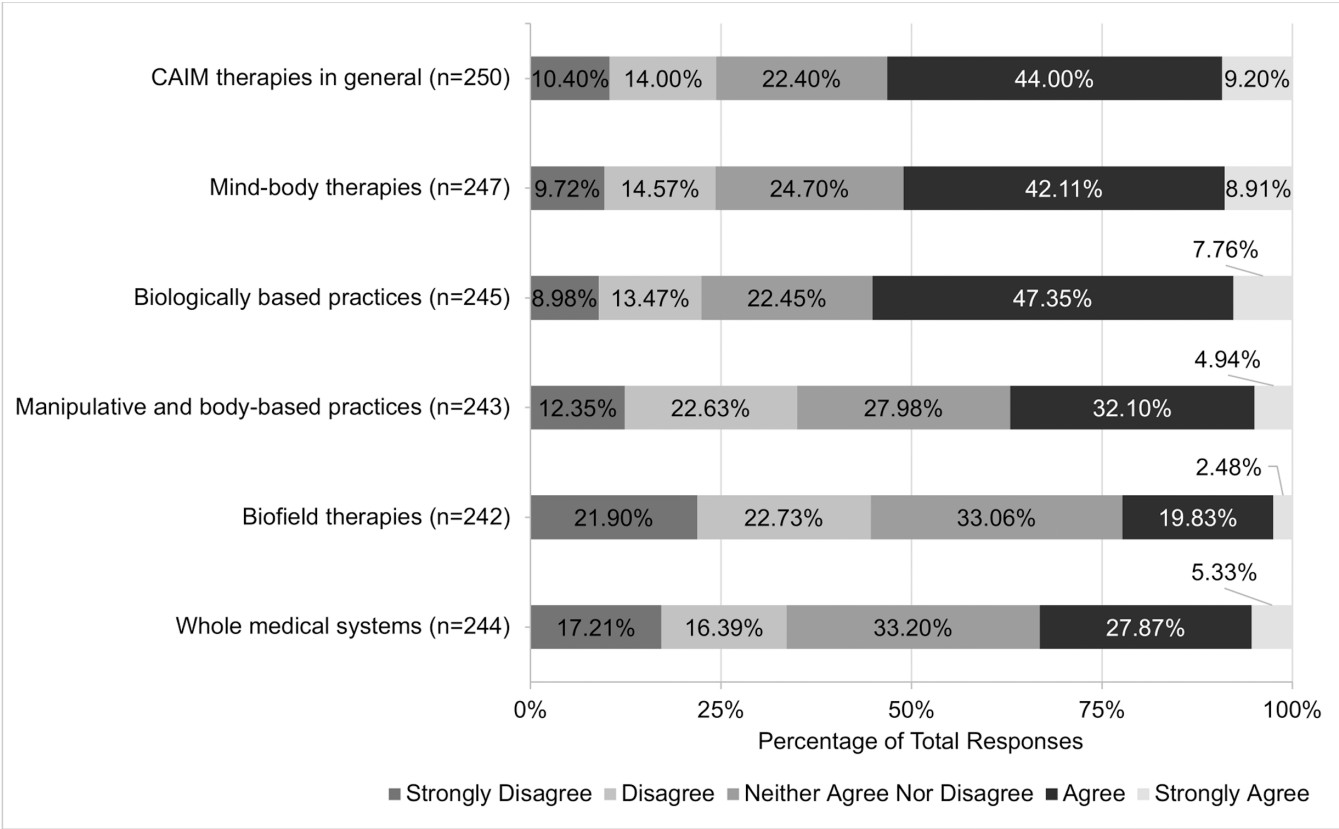

**Fig 9. Participants' Agreement Expressed Towards Clinicians Receiving Formal Education on Each Complementary, Alternative, and Integrative Medicine (CAIM) Category.**

## Strengths and limitations

One strength of this study is the utilization of a large-scale, international, cross-sectional survey, which resulted in a large sample comprised of individuals with varied perspectives regarding CAIM. This survey acquired responses from researchers and clinicians with a range of professional roles and experiences, which improved its generalizability to the cardiology field in general.

With regards to limitations, the survey was exclusively administered in English, thus our findings may be less generalizable to the non-English speaking cardiology researchers and clinicians. The largest proportion of respondents were from Europe and the Americas, potentially reducing the generalizability of the findings in other world regions [81]. Several factors could have contributed to this geographic inclination, including restricted access to SurveyMonkey in certain countries, English-language barriers, and fewer publishing opportunities in underrepresented regions. This limitation is relevant because prior research has found that personal characteristics associated with lived experiences (e.g., age, sex, prior exposure, cultural background, and education) and the geographic location of individuals have been seen to influence the use and perceptions of CAIM therapies [82,83]. Furthermore, the categorization of CAIM into five subcategories within the survey may have oversimplified the nuanced efficacy, perceptions, and safety of each modality, leading participants to provide generalized opinions of these therapies. This issue was evident in the thematic analysis, where many participants expressed concerns about the broad categorization of CAIM, highlighting potential flaws in accurately assessing their perceptions. Lastly, bias may have been introduced due to the nature of the study. For instance, responses within the

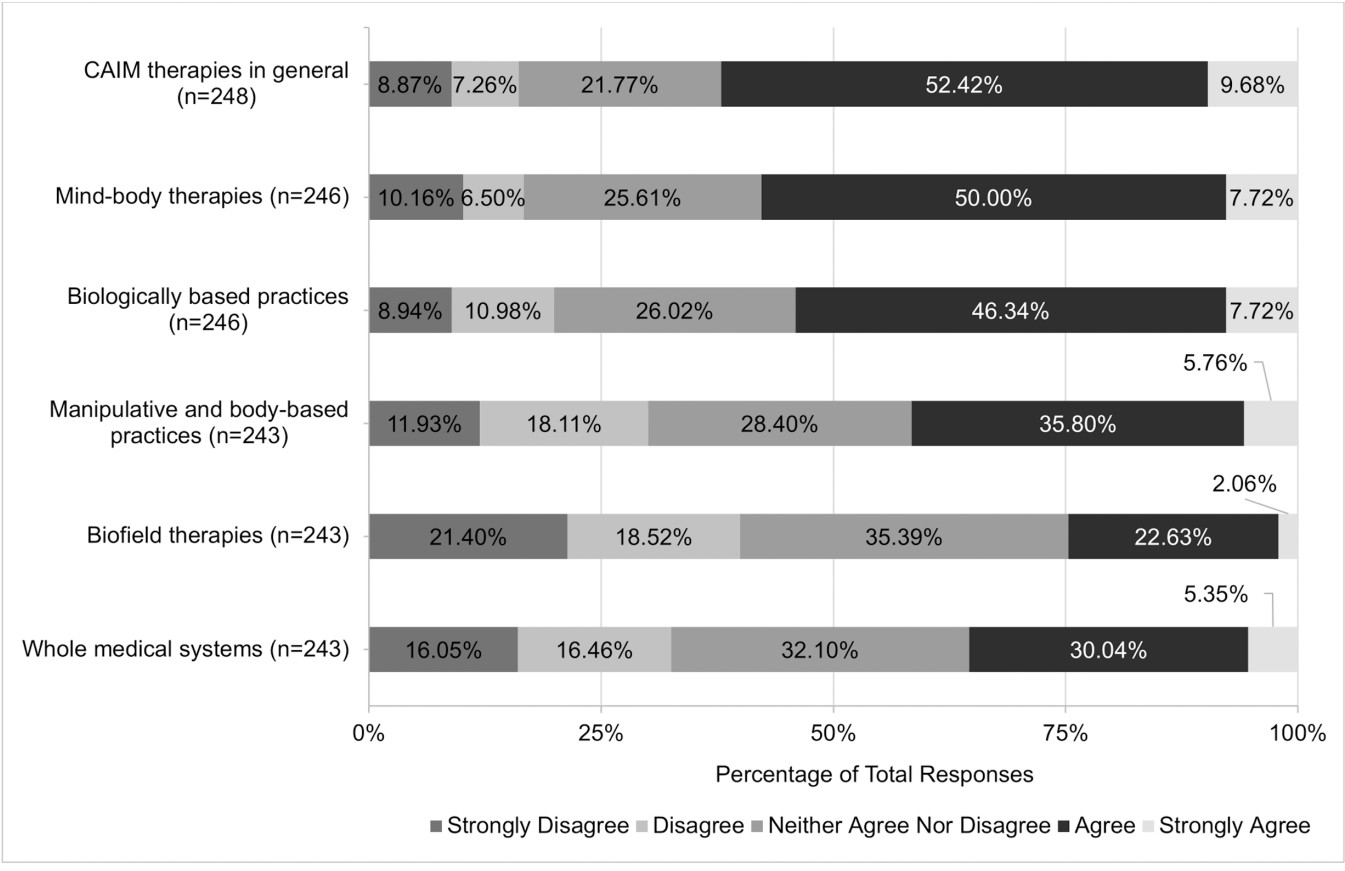

**Fig 10. Participants' Agreement Expressed Towards Clinicians Receiving Supplementary Education on Each Complementary, Alternative, and Integrative Medicine (CAIM) Category.**

present analysis could be susceptible to response bias, as differences between responders and non-responders to the online survey could impact the representativeness of the sample [84]. Alternatively, a difference in response between survey respondents and non-responders, otherwise known as non-response bias, may have also been introduced because researchers and clinicians with an interest in CAIM are more likely to partake in the survey [84]. Moreover, the recruitment strategy may have introduced sampling bias. The contact information (name and email address of the corresponding author) of potential participants was extracted from research articles published in cardiology journals; understandably this resulted in proportionately more researchers (38.19%, n = 118) who participated, followed by those who are both researchers and clinicians (55.99%, n = 173), while the least were respondents who are solely clinicians (5.83%, n = 18). Participants may also have been subject to recall bias, where responses are influenced by the varying accuracy and knowledge with which individuals remember their own experiences due to the self-reporting nature of the survey [84].

## Conclusion

This study investigated the practice and perceptions of researchers and clinicians within the field of cardiology. The present analysis highlights the necessity for further research on CAIM and emphasizes the importance of providing both formal and informal education and training for cardiology professionals in this field. Among the CAIM subcategories, mind-body therapies and biologically based practices were the most favored, and biofield therapies were the least favored by

**Table 2. Summary of thematic analysis findings.**

| Theme | Representative Quotes |
|---|---|
| Need for more rigorous research and resources to inform decision making about CAIM | *"Alternative medicine practices whose benefits have not been scientifically proven should not, in my opinion, be applied to patients. The practice must be scientifically proven to be beneficial."*<br>*"I have seen a number of CAIM therapies work in a number of patients, even though I don't know the mechanism. The lack of consistency in reporting and evidence, however, keeps me from prescribing it myself. I hope more research on CAIM will show us who can be treated with what CAIM, because I see the need in a number of my patients for non-traditional therapies, whether through their own more holistic beliefs or through an active distrust of 'authorities' in general (or combination of such reasons)."* |
| CAIM products can be harmful due to misuse/misinformation | *"I am happy to recommend some CAIM where there is some evidence of benefit and evidence of lack of harm, but am concerned when patients use the se [sic] as an alternative to well tested and proven therapies. They are better used in a complementary fashion."*<br>*"Some can be harmful and I think that takes the priority of first do no harm, also very money making"* |
| Concern with CAIM categories | *"I think chiropractic medicine should be separated from other reflexology/massage, because I think massage and other forms are great, but chiropractic medicine is much more dangerous."*<br>*"[It is] difficult to answer question in some categories due to multiple items listed. Chiropractic is different from others in the category, dangerous in many situations. Similarly, homeopathy and naturopathy are different from others in the category in that I would recommend to avoid them but others in the category would be acceptable."* |
| CAIM is appealing to patients | *"In Mexico exit [sic] a lot of CAIM. However, there is not scientific support for their use at all. Moreover, patients prefer these therapies over professional medical advice at public hospitals."*<br>*"Not clear they are effective in any quantifiable way, but most patients seem to feel better. It's a way of getting patients to invest in their well being"* |
| Polar or personal opinions/experiences (pro vs. anti | *"It's largely unscientific, unfalsifiable bullshit hindered by practitioners unwilling to risk compromising their profitable schemes by subjecting them to scientific scrutiny."*<br>*"I have seen positive effects of mind/body treatment on hypertension and PVC's' palpitations."* |
| CAIM in the clinical and/or academic field of cardiology | *"I think there is potential in many of these therapies for more general health and lifestyle choices (not necessary Cariology-specific [sic]), but theres [sic] not enough valid standardisation of information/guidelines"*<br>*"There is potential for harm in certain high-risk sub-populations (end-stage heart failure, pulmonary vascular disease, etc) with complex medical conditions and a wide range of medications, in whom certain CAIM therapies (vitamins, supplements, herbs) can interact with medications and medical condition adversely"* |

cardiology researchers and clinicians. With the importance of prevention and lifestyle modifications for cardiac health, along with the increasing number of patients turning to CAIM, it is essential for cardiology professionals to have access to research and education tailored to meet their patients' needs. This study serves as a foundation for such endeavors.

## Supporting information

**S1 Checklist. STROBE Checklist Aug2224.**
(DOCX)

## Acknowledgments

We acknowledge support from the Open Access Publication Fund of the University of Tübingen

## Author contributions

**Conceptualization:** Jeremy Y. Ng, Holger Cramer.

**Data curation:** Jeremy Y. Ng, Mehvish Masood, Sivany Kathir.

**Formal analysis:** Jeremy Y. Ng, Mehvish Masood, Sivany Kathir.

**Investigation:** Jeremy Y. Ng, Mehvish Masood, Sivany Kathir, Holger Cramer.

**Methodology:** Jeremy Y. Ng, Holger Cramer.

**Project administration:** Jeremy Y. Ng, Holger Cramer.

**Software:** Jeremy Y. Ng.

**Supervision:** Jeremy Y. Ng, Holger Cramer.

**Writing – original draft:** Jeremy Y. Ng, Mehvish Masood, Sivany Kathir.

**Writing – review & editing:** Jeremy Y. Ng, Mehvish Masood, Sivany Kathir, Holger Cramer.

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
