## [Decision Letter · Decision Letter 0]

5 Feb 2025

PONE-D-24-36458Perceptions of Complementary, Alternative, and Integrative Medicine: A Global Cross-Sectional Survey of Cardiology Researchers and CliniciansPLOS ONE

Dear Dr. Ng,

Thank you for submitting your manuscript to PLOS ONE. After careful consideration, we feel that it has merit but does not fully meet PLOS ONE’s publication criteria as it currently stands. Therefore, we invite you to submit a revised version of the manuscript that addresses the points raised during the review process.

We look forward to receiving your revised manuscript.

Kind regards,

Satish G Patil, PhD

Academic Editor

PLOS ONE

Journal Requirements:

Additional Editor Comments:

I request authors to respond to the following Major comments:

1. Why under alternative 'Whole Medical System' only Homoeopathy is considered? Why not other alternative systems such as Ayurveda, Chinese Medicine or Unani etc? Please clarify

2. And why the Survey was only done in English Language, when many alternative system are practiced in English non-speaking countries?

3. Whether, the Cardiologists who responded to the survey are aware OR know the concept/Principles of practice of CAIM? if they don't know, than whether their responses are significant.

4. In the introduction section, the second page is less relevant to the objectives of the research (CAIM and Cardiologists).

Few minor corrections:

5. Table-1: I think the age is given in years, but 'Years' is not mentioned.

6. Fig-3: in the caption, is it should be 'Benefits to Participants' correct it.

7. Similarly in Fig-4, Correct the caption.

Reviewers' comments:

Reviewer's Responses to Questions

**Comments to the Author**

1. Is the manuscript technically sound, and do the data support the conclusions?

Reviewer #1: Partly

2. Has the statistical analysis been performed appropriately and rigorously?

Reviewer #1: Yes

3. Have the authors made all data underlying the findings in their manuscript fully available?

Reviewer #1: Yes

4. Is the manuscript presented in an intelligible fashion and written in standard English?

Reviewer #1: Yes

5. Review Comments to the Author

Reviewer #1: PLOS One review

- The relationship between your recruitment strategy and your target population is a bit unclear. How do you know the survey was sent to clinicians? If you intended to recruit clinicians, is this the best way to go about it? The recruitment strategy appears to prioritise researchers. Equally, when reviewing the Methods, I could not see any reference to the need for participants to be either a researcher or a clinician to be eligible to participate.

- Are these surgeries expensive and inaccessible everywhere? Not all health systems operate using the same costing methods. It would be good to be clear about where the evidence for this statement is coming from.

- (INTRODUCTION) Is homeopathy the best example of a whole medical system here? I would have thought Traditional Chinese medicine, Ayurveda or Naturopathy would be better here as they are more widely used internationally and have more documented practice principles and philosophies. I don't think Acupuncture is a whole medical system either, as is stated in the Methods section.

- Is there ANY previous research examining this topic? You say 'limited' but have not provided an overview of anything prior. It would be good to have these background data so we can be clear about what gap this study is filling.

- Why do you think there were so few respondents from other world regions?

- I am intrigued - and a little skeptical of - the value of evaluating perceived safety and effectiveness of 'CAIM' by such broad categories. Given the evidence varies so much between specific interventions and for specific conditions.

- How do you differentiate between 'counselling' and 'recommending'? How confident are you that the participants interpretation of these words aligned with your intended meaning?

- It would be good to include some example quotes for the thematic analysis. These could simply be included in a table rather than in the text if word count is an issue.

- Is it not also worth mentioning in the Discussion - in response to the point about the perceived efficacy of biologically-based factors - that while there is evidence of safety issues through drug interactions, there is strong evidence for the effectiveness of some biologicals. Is it not also possible that the reason for the low rating among study participants is that they are basing their judgement on their own definition of what needs to happen when something 'works'? Or other reasons? This is a very superficial interpretation of the finding.

I also find the rate of use/disclosure reported by participants interesting given that there is such a low rate of disclosure of CM use to medical doctors. This may be worth discussing further.

6. PLOS authors have the option to publish the peer review history of their article (what does this mean? ). If published, this will include your full peer review and any attached files.

**Do you want your identity to be public for this peer review?** For information about this choice, including consent withdrawal, please see our Privacy Policy .

Reviewer #1: No

---

## [Author Response · Author response to Decision Letter 1]

13 Mar 2025

Response to Reviewers

Dear Editor:

Thank you for your email. I am pleased to hear that you believe that our manuscript will have the potential to be published in PLOS ONE following requested revisions.

As per your request, please find our response to the peer-reviewers’ comments with every change outlined point by point below:

● We kindly thank the editor for providing their feedback on our manuscript.

Journal Requirements:

● We have ensured that our manuscript adheres to PLOS ONE's style requirements, including file naming conventions, formatting, and structure.

● The ethics statement was already placed in the Methods section, and we have confirmed that it does not appear in any other section of the manuscript.

● We have no supporting information files.

Additional Editor Comments:

I request authors to respond to the following Major comments:

1. Why under alternative 'Whole Medical System' only Homoeopathy is considered? Why not other alternative systems such as Ayurveda, Chinese Medicine or Unani etc? Please clarify

● We acknowledge that ‘Whole Medical System’ is not limited to Homeopathy and includes other alternative systems such as Ayurveda, Chinese Medicine or Unani. The introduction has been updated to reflect this change: “This involves, but is not limited to, the use of mind-body therapies (e.g., meditation, biofeedback, yoga), biologically based practices (e.g., vitamins, botanicals, special foods and diets), manipulative and body-based practices (e.g., massages, reflexology, chiropractic therapy), and whole medical systems (e.g., traditional Chinese medicine, Ayurveda, naturopathic medicine).”

2. And why the Survey was only done in English Language, when many alternative system are practiced in English non-speaking countries?

● The survey was conducted in English due to practical and feasibility constraints, as this was an unfunded study. Expanding the survey to multiple languages would have required additional resources, such as translators and validation processes for survey materials. Additionally, English is the predominant language of scientific research, with approximately 98% of scientific publications written in English (Ramírez-Castañeda, 2020). Given that most cardiology researchers and clinicians publish in English, we considered it the most appropriate language for our study.

● This limitation is mentioned in the Strengths and Limitations section of the discussion: “First, the survey was exclusively administered in English.”

Ramírez-Castañeda, V. (2020). Disadvantages in preparing and publishing scientific papers caused by the dominance of the English language in science: The case of Colombian researchers in biological sciences. PloS one, 15(9), e0238372.

3. Whether, the Cardiologists who responded to the survey are aware OR know the concept/Principles of practice of CAIM? if they don't know, than whether their responses are significant.

● We did not require participants to have any prior knowledge of CAIM, nor did we exclude those with limited/no familiarity. Our objective was to capture a broad spectrum of perspectives, from those well-versed in CAIM to those who may have minimal exposure. This approach allows us to better understand how CAIM is perceived within the cardiology community and whether there are gaps in knowledge that may impact its integration into clinical practice.

● We added the following to the methods to highlight this concept: “Respondents were not required to have any background or understanding of CAIM because this study intended to capture how CAIM is perceived within the general cardiology community and whether there are gaps in knowledge that may impact its integration into research/clinical practice.”

4. In the introduction section, the second page is less relevant to the objectives of the research (CAIM and Cardiologists).

● We have reviewed the introduction and have removed the following section: “A cross-sectional study investigating the knowledge and use of CAM among CVD patients (n = 90) found that 63.3% of participants reported using CAM.18 A survey given to pediatric patients (or their parents/guardians) in two cardiology units found that 59.1% of patients had utilized CAM at one point in their lives, with multivitamins and massages being the most common products and practices reported, respectively.30” Furthermore, other minor edits have been made to condense that section of the introduction.

Few minor corrections:

5. Table-1: I think the age is given in years, but 'Years' is not mentioned.

● We have updated Table 1 to explicitly mention that age is given in years for clarity.

6. Fig-3: in the caption, is it should be 'Benefits to Participants' correct it.

● We have corrected the caption to read "Benefits Participant Perceive as Associated with Complementary, Alternative, and Integrative Medicine (CAIM)”

7. Similarly in Fig-4, Correct the caption.

● We have corrected the caption to read "Challenges Participants Perceive as Associated with Complementary, Alternative, and Integrative Medicine (CAIM)”.

Reviewers' comments:

Reviewer's Responses to Questions

Comments to the Author

1. Is the manuscript technically sound, and do the data support the conclusions?

Reviewer #1: Partly

2. Has the statistical analysis been performed appropriately and rigorously?

Reviewer #1: Yes

3. Have the authors made all data underlying the findings in their manuscript fully available?

Reviewer #1: Yes

4. Is the manuscript presented in an intelligible fashion and written in standard English?

Reviewer #1: Yes

5. Review Comments to the Author

Reviewer #1: PLOS One review

- The relationship between your recruitment strategy and your target population is a bit unclear. How do you know the survey was sent to clinicians? If you intended to recruit clinicians, is this the best way to go about it? The recruitment strategy appears to prioritise researchers. Equally, when reviewing the Methods, I could not see any reference to the need for participants to be either a researcher or a clinician to be eligible to participate.

● While the survey was sent to individuals who have published in cardiology journals, participation was limited to those who identified as cardiology researchers or clinicians.

o To be eligible for the study, participants were required to answer ‘yes’ to the following screening question: “Do you self-identify as a researcher or clinician within the field of cardiology? [NOTE: Survey will end if respondent answers “No” to question #2. Survey logic will be applied so that those who are only researchers will see a certain subset of questions, those who are clinicians will see a certain subset of questions, and those who are both will see all questions. This is all specified in the questions below.]”

o The survey’s screening question can be accessed here: https://doi.org/10.17605/OSF.IO/MQBW6

o We have revised the Methods section to explicitly state this: “To be eligible, participants needed to be able to read and write in English and self-identify as a researcher and/or clinician within the field of cardiology.”

● We acknowledge that the sampling strategy may introduce bias into the study. The sampling strategy was chosen in part due to practical and feasibility constraints, as this was an unfunded study. However, as this is the first survey of its kind, our goal was to establish an initial understanding of how cardiology researchers and clinicians perceive CAIM. Future research can build upon this by exploring perceptions of clinicians and researchers.

o The following was added in the limitation section: “Moreover, the recruitment strategy may have introduced sampling bias. The contact information (name and email address of the corresponding author) of potential participants was extracted from research articles published in cardiology journals; understandably this resulted in proportionately more researchers (38.19%, n=118) who participated, followed by those who are both researchers and clinicians (55.99%, n=173), while the least were respondents who are solely clinicians (5.83%, n=18).”

- Are these surgeries expensive and inaccessible everywhere? Not all health systems operate using the same costing methods. It would be good to be clear about where the evidence for this statement is coming from.

● We acknowledge that healthcare costs and accessibility vary by region. We have revised the text to specify where the cost burden is most notable, drawing from the studies that were cited on the U.S. and China: “Similarly, surgeries such as coronary artery bypass grafts and bariatric surgery are known to be both expensive and inaccessible, especially in countries including the United States and China”

- (INTRODUCTION) Is homeopathy the best example of a whole medical system here? I would have thought Traditional Chinese medicine, Ayurveda or Naturopathy would be better here as they are more widely used internationally and have more documented practice principles and philosophies. I don't think Acupuncture is a whole medical system either, as is stated in the Methods section.

● We acknowledge that ‘Whole Medical System’ is not limited to Homeopathy and includes other alternative systems such as Ayurveda, Chinese Medicine or Unani. The introduction has been updated to reflect this change: “This involves, but is not limited to, the use of mind-body therapies (e.g., meditation, biofeedback, yoga), biologically based practices (e.g., vitamins, botanicals, special foods and diets), manipulative and body-based practices (e.g., massages, reflexology, chiropractic therapy), and whole medical systems (e.g., traditional Chinese medicine, Ayurveda, naturopathic medicine).”

- Is there ANY previous research examining this topic? You say 'limited' but have not provided an overview of anything prior. It would be good to have these background data so we can be clear about what gap this study is filling.

● To the best of our knowledge, research has not been conducted on this topic previously. The introduction has been edited accordingly: “To the best of our knowledge, research has not been conducted on the perspectives of CAIM among cardiology researchers and physicians specifically, with previous literature focusing on CAIM perspectives in general [37], or focused specifically on perspectives of patients with cardiovascular disease[38,39].”

- Why do you think there were so few respondents from other world regions?

● Several factors may have contributed to the low number of respondents from world regions outside the Americas and Europe. These include language barriers, as the survey was only available in English, restricted access to SurveyMonkey in certain countries, and less publishing opportunities in underrepresented regions. Since we recruited participants based on authorship of cardiology publications, this naturally influenced the regional distribution of respondents.

● We have added this discussion to the Limitations section: “Second, a predominant proportion of respondents were from Europe and the Americas, potentially reducing the generalizability of the findings [79]. Several factors could have contributed to this geographic inclination, including restricted access to SurveyMonkey in certain countries, English-language barriers, and fewer publishing opportunities in underrepresented regions.”

- I am intrigued - and a little skeptical of - the value of evaluating perceived safety and effectiveness of 'CAIM' by such broad categories. Given the evidence varies so much between specific interventions and for specific conditions.

● We acknowledge that evidence varies between specific interventions and conditions. However, the broad categories allow us to capture the general perspectives of cardiology clinicians and researchers. Given that patients use CAIM regardless of clinician recommendations, we believe it is still valuable to assess perceptions at a general level, especially as this is the first study of its kind on this topic. Certainly, future work can aim to explore more specific interventions within the context of CAIM.

- How do you differentiate between 'counselling' and 'recommending'? How confident are you that the participants interpretation of these words aligned with your intended meaning?

● We define counseling as providing informed education and discussing risks/benefits, whereas recommending refers to making suggestions about course of action. While it won’t be possible to know how participants interpreted these words (the same could be said for any question in any cross-sectional survey), it is worth mentioning that this survey was pilot-tested by a group of researchers familiar with this topic area.

- It would be good to include some example quotes for the thematic analysis. These could simply be included in a table rather than in the text if word count is an issue.

● We have added Table 2, which includes representative quotes from each theme/subtheme to enhance the transparency of our thematic analysis.

- Is it not also worth mentioning in the Discussion - in response to the point about the perceived efficacy of biologically-based factors - that while there is evidence of safety issues through drug interactions, there is strong evidence for the effectiveness of some biologicals. Is it not also possible that the reason for the low rating among study participants is that they are basing their judgement on their own definition of what needs to happen when something 'works'

---

## [Decision Letter · Decision Letter 1]

21 Mar 2025

Perceptions of Complementary, Alternative, and Integrative Medicine: A Global Cross-Sectional Survey of Cardiology Researchers and Clinicians

PONE-D-24-36458R1

Dear Dr. Ng,

We’re pleased to inform you that your manuscript has been judged scientifically suitable for publication and will be formally accepted for publication once it meets all outstanding technical requirements.

Kind regards,

Satish G Patil, PhD

Academic Editor

PLOS ONE

Additional Editor Comments (optional):

Reviewers' comments:

Reviewer's Responses to Questions

**Comments to the Author**

1. If the authors have adequately addressed your comments raised in a previous round of review and you feel that this manuscript is now acceptable for publication, you may indicate that here to bypass the “Comments to the Author” section, enter your conflict of interest statement in the “Confidential to Editor” section, and submit your "Accept" recommendation.

Reviewer #1: All comments have been addressed

2. Is the manuscript technically sound, and do the data support the conclusions?

Reviewer #1: Yes

3. Has the statistical analysis been performed appropriately and rigorously?

Reviewer #1: Yes

4. Have the authors made all data underlying the findings in their manuscript fully available?

Reviewer #1: Yes

5. Is the manuscript presented in an intelligible fashion and written in standard English?

Reviewer #1: Yes

6. Review Comments to the Author

Reviewer #1: Thank you for addressing all of my points. While there is space for a more detailed discussion on some of the points I raised, I respect the authors decisions with regards to what they include as discussion in their manuscript.

7. PLOS authors have the option to publish the peer review history of their article (what does this mean? ). If published, this will include your full peer review and any attached files.

**Do you want your identity to be public for this peer review?** For information about this choice, including consent withdrawal, please see our Privacy Policy .

Reviewer #1: No

---

## [Editor Report · Acceptance letter]

PONE-D-24-36458R1

PLOS ONE

Dear Dr. Ng,

I'm pleased to inform you that your manuscript has been deemed suitable for publication in PLOS ONE. Congratulations! Your manuscript is now being handed over to our production team.

Kind regards,

on behalf of

Dr. Satish G Patil

Academic Editor

PLOS ONE